# Bimanual Movement Characteristics and Real-World Performance Following Hand–Arm Bimanual Intensive Therapy in Children with Unilateral Cerebral Palsy

**DOI:** 10.3390/bs13080681

**Published:** 2023-08-13

**Authors:** Shailesh S. Gardas, Christine Lysaght, Amy Gross McMillan, Shailesh Kantak, John D. Willson, Charity G. Patterson, Swati M. Surkar

**Affiliations:** 1Department of Physical Therapy, East Carolina University, Greenville, NC 27834, USA; gardass21@students.ecu.edu (S.S.G.); lysaghtc@ecu.edu (C.L.); grossmcmillana@ecu.edu (A.G.M.); willsonj@ecu.edu (J.D.W.); 2Moss Rehabilitation Research Institute, Elkins Park, PA 19027, USA; 3Department of Physical Therapy, Arcadia University, Glenside, PA 19038, USA; 4Department of Physical Therapy and School of Health and Rehabilitation Sciences Data Center, University of Pittsburgh, Pittsburgh, PA 15260, USA; cgp22@pitt.edu

**Keywords:** training intensity, bimanual coordination, real-world activity, actigraphs, upper extremity

## Abstract

The purpose of this study was to quantify characteristics of bimanual movement intensity during 30 h of hand–arm bimanual intensive therapy (HABIT) and bimanual performance (activities and participation) in real-world settings using accelerometers in children with unilateral cerebral palsy (UCP). Twenty-five children with UCP participated in a 30 h HABIT program. Data were collected from bilateral wrist-worn accelerometers during 30 h of HABIT to quantify the movement intensity and three days pre- and post-HABIT to assess real-world performance gains. Movement intensity and performance gains were measured using six standard accelerometer-derived variables. Bimanual capacity (body function and activities) was assessed using standardized hand function tests. We found that accelerometer variables increased significantly during HABIT, indicating increased bimanual symmetry and intensity. Post-HABIT, children demonstrated significant improvements in all accelerometer metrics, reflecting real-world performance gains. Children also achieved significant and clinically relevant changes in hand capacity following HABIT. Therefore, our findings suggest that accelerometers can objectively quantify bimanual movement intensity during HABIT. Moreover, HABIT enhances hand function as well as activities and participation in real-world situations in children with UCP.

## 1. Introduction

Unilateral cerebral palsy (UCP) is a leading cause of childhood disability [1]. Children with UCP have difficulty with bimanual coordination, which further restricts the child’s independence in daily activities and impairs quality of life [2]. Hand–arm bimanual intensive therapy (HABIT) is a well-established intervention to improve hand function and bimanual coordination in children with UCP [3]. Traditionally, the intensity of HABIT has been quantified as the number of hours of therapy [4,5]. Despite the reported improvements in upper extremity (UE) function, discrepancies exist in the intensity of training protocols [6,7]. Evidence indicates that greater intensities in terms of hours result in larger gains in UE motor outcomes [4,8]. However, time is recognized as a proxy measure of training intensity since it does not reveal specific information about the goal-directed UE or bimanual movements occurring during training [9]. Hence, it is crucial to develop objective methods to quantify the intensity of HABIT in terms of bimanual movement characteristics that would provide insights into bimanual training intensity.

Accelerometers have been accepted as a valid tool to objectively capture UE movement characteristics in the real-world environment [10,11,12]. They measure accelerations of UE movements along the predefined axes in gravitational units called activity counts. The gold standard methods to monitor quality and quantity of UE use, such as 3D kinematic [13] and human-observed coding [14] of video recordings, can be time-consuming and expensive to quantify the intensity of HABIT. Accelerometer metrics overcome these problems and provide a convenient option to measure movement characteristics during intensive therapy as well as in the real-world environment, thus capturing the activities and participation domains per the International Classification of Functioning, Disability, and Health—Children and Youth Version (ICF-CY) [15]. Acceleration metrics that reflect UE movement characteristics are broadly classified into three categories [16]. The first category indicates the relative contributions of the affected and less affected UE movements using use ratio, magnitude ratio, and bilateral magnitude [16]. The second category comprises characteristics specific to the accelerations of the affected extremity through median acceleration and acceleration variability. The third category provides the number of accelerations using activity counts for both extremities [16]. Our novel approach capitalized on these three categories to quantify the characteristics of a 30 h (intensity) HABIT. An understanding of bimanual movement characteristics can provide clinically meaningful information regarding the contribution of the affected vs. less affected extremity to bimanual activities during HABIT and thereby guide clinical dosing criteria and capture gains in bimanual performance in real-world activities.

Performance refers to what a person actually does in a real-world environment, whereas capacity refers to what a person can do in a controlled environment such as a clinic [17]. The International Classification of Functioning, Disability, and Health—Children and Youth Version (ICF-CY) by the World Health Organization, provides a clear differentiation between performance and capacity within the domains of activities and participation [15]. Activity refers to the execution of a task or action by an individual, whereas participation is involvement in a life situation. Performance qualifier, according to ICF-CY, refers to what a person does in a current (real-world or lived experience) environment, signifying participation, whereas capacity refers to what a person can do in a controlled (standard) environment such as in a clinic, signifying activity [17]. Traditionally, studies examining the efficacy of HABIT have assessed changes in UE capacity using standardized clinical tests such as the Assisting Hand Assessment (AHA), Box and Block Test (BBT), Nine-Hole Peg Test (NHPT), Jebsen Hand Function Test (JHFT), etc., that primarily capture body function and activity per ICF-CY [18]. UE performance (activity and participation) has been assessed using self-reported measures such as the Canadian Occupational Performance Measure, the Pediatric Evaluation and Disability Inventory, and ABILHAND [18]. Collectively, these studies indicate that HABIT improves UE capacity as well as the performance of children with UCP. However, self-reported measures are prone to subjective and social desirability biases, which raise questions about whether in-clinic improvements are indicative of changes in real-world bimanual activities and participation [19]. Recent evidence supports this conjecture since discrepancies between parents’ perceptions of their child’s performance using self-reported measures and therapists’ assessments of capacity have been reported [20]. Furthermore, changes in standardized assessments may not translate to improvements in the affected UE use in daily life when assessed with accelerometers in children with UCP [21,22] as well as the adult stroke population [23]. Therefore, capturing the performance of the affected UE during daily bimanual activities using accelerometers is crucial to elucidate whether improvements in capacity with HABIT translate to gains in performance in real-world bimanual activities and participation in children with UCP.

In the last decade, accelerometers have been primarily used to assess UE performance after intervention in adult stroke survivors [24,25] and to detect motor asymmetries in children with UCP [26]. Only a few studies have used accelerometers to monitor UE gains in children with UCP following intensive therapy such as constraint-induced movement therapy (CIMT) [21,27]. Collectively, the results of the studies in adults with stroke [16] as well as in children with UCP [21,27] indicate that despite improvements in UE capacity, UE performance in daily life showed little to no improvement. Despite the excellent capacity of accelerometers to capture activity counts and movement characteristics in a real-world environment, none of the studies have utilized accelerometers to measure real-world bimanual performance post-HABIT.

Therefore, the primary purpose of this novel study was to objectively quantify the characteristics of bimanual movements during 30 h (intensity) of HABIT using bilateral wrist-worn accelerometers. The second purpose was to examine the gains in real-world bimanual performance following HABIT using accelerometer-derived variables reflecting activities and participation. We hypothesize that accelerometers will accurately capture the UE movement characteristics reflecting bimanual use during HABIT. Furthermore, 30 h of HABIT will enhance children’s affected UE contributions to the performance of real-world bimanual activities.

## 2. Materials and Methods

### 2.1. Study Design and Setting

This study is an ancillary analysis [28] of a double-blind, randomized controlled trial (NCT05355883). It was a prospective pre- and post-training study conducted at the Pediatric Assessment and Rehabilitation Laboratory (PeARL) at East Carolina University (ECU), NC. The University and Medical Center Institutional Review Board, ECU, approved the study. We obtained parental consent and child assent. The assessors were blinded to the pre- and post-testing assessments. The study was conducted between November 2021 and January 2023.

### 2.2. Participants

Twenty-five children with UCP, ages 6–16 years (mean age = 11.20 ± 3.59 years), and Manual Ability Classification System levels I–III participated in this study. Figure 1 shows the CONSORT diagram, describing the flow of participants through the study, withdrawals, and inclusion in analyses.

Children with other neuromotor disabilities, cognitive and communication deficits, cardiorespiratory dysfunctions, metabolic disorders, neoplasms, and a history of botulinum neurotoxin injections on the affected UE in the past 6 months were excluded. Table 1 describes further details about participant characteristics.

### 2.3. Procedures

#### 2.3.1. Hand–Arm Bimanual Intensive Therapy (HABIT) Protocol

HABIT is a well-established intervention shown to improve bimanual coordination in children with UCP [3]. We administered HABIT in a camp-based setting with a pre-determined duration of a total of 30 h of structured, task-specific, bimanual activities, six hours per day for five consecutive days. The therapy included age-appropriate bimanual gross and fine motor tasks in a playful context (please see Appendix A). The child-to-interventionist ratio (trained physical and occupational therapy students) was 1:4. Individualized therapy goals were formulated based on the pre-training behavioral hand function tests as well as parent–child identified bimanual goals. Interventionists progressively increased the complexity of bimanual activities. The task demands were graded based on the task performance to allow the child to complete the task successfully. Children were encouraged to use the affected and the less affected UE in a coordinated manner. Positive reinforcement and knowledge of performance were provided to motivate and reinforce desired goal-directed activities. Emphasis was placed on different roles of the affected UE, such as stabilizer, manipulator, and assistor, depending on the child’s ability and task goal. Sessions comprised whole-task and part-task practice. Throughout the HABIT, three licensed physical therapists supervised the interventionists to ensure the fidelity of therapy. Activities performed by the children were documented by the interventionists [7].

#### 2.3.2. Accelerometry Methodology

Bilateral wrist-worn accelerometers (Actigraphy GT9X Link, Pensacola, FL, USA) were used to quantify the movement characteristics during the five days of HABIT and the performance (real-world UE activity) gains post-HABIT. Actigraph GT9X Link measures accelerations in activity counts along three axes, with one count equaling 0.001664 g [29]. Accelerometer data were sampled at 30 Hz, and activity counts were binned into 1 s epochs for each axis using ActiLifeTM 6 software. Data was then processed in MATLAB (Mathworks Inc, Natick, MA, USA) using custom-written software developed by Lang et al. [29]. To determine movement characteristics during HABIT days, children wore accelerometers for 5 days of HABIT (30 h total wear time). To measure UE performance gains, children wore accelerometers for three consecutive days pre- and post-HABIT during their daily activities, which included home, school, and play. This approach was designed to capture bimanual activities throughout the day in a natural, real-world environment. Moreover, the three-day accelerometer wearing time pre- and post-HABIT was chosen since it produces a reliable estimate of performance in children with CP [30]. Detailed instructions were provided to both parents and children on the proper wearing and usage of accelerometers. Specifically, they were instructed to keep the Actigraphs on during waking hours, but to remove them while bathing or engaging in water-related activities.

### 2.4. Outcome Measures

#### 2.4.1. Bimanual Movement Intensity Characteristics and Performance Measures

##### *Accelerometer-derived metrics [16,29]: Activity and Participation Domains* *of ICF-CY*

We quantified the bimanual movement characteristics using six standard accelerometry derived variables: (1) use ratio (UR), (2) magnitude ratio (MR), (3) bilateral magnitude (BM), (4) median acceleration (MA), (5) acceleration variability (AV), and (6) affected UE activity counts (AAC). Changes in UE performance pre- and post-HABIT reflecting real-world performance (activity and participation) were assessed using (1) UR, (2) MR, (3) BM, (4) MA, and (5) AV [15]. Figure 2 explains the accelerometry-derived variables.

(1)The use ratio reflects the contribution of the affected UE relative to the less affected UE and is calculated as the ratio of the active duration of the affected arm to that of the less affected arm. The UR value ranges between 0 and 1. A value close to or equal to 1 indicates symmetrical use of the extremities, whereas a value closer to zero indicates less affected UE use.(2)The magnitude ratio is the ratio of acceleration magnitude (range of movement) of both UEs and is calculated by dividing the acceleration magnitude of the affected and the less affected UE. The value of MR ranges from −7 to +7. A value closer to 0 indicates equal contributions from both UEs; positive values indicate greater movement magnitude of the affected UE, and negative values indicate greater movement magnitude of the less affected UE.(3)Bilateral magnitude reflects the magnitude of accelerations across both UEs and is calculated by summing the smoothed vector magnitudes of both UEs for each second of activity. Zero indicates no activity, and an increasing value indicates greater magnitudes of bilateral UE activity.(4)Median acceleration and acceleration variability are variables that reflect movement characteristics considering only the affected UE. The median acceleration represents the acceleration of the affected UE magnitude over the entire wear time.(5)Acceleration variability is the variance of the mean acceleration and represents the average distance of the affected UE accelerations from the mean acceleration. A higher score for both of these variables indicates better overall UE movement and variability, respectively [16,21].(6)Affected extremity activity counts quantify the number of affected extremity accelerations (activity counts) during therapy.

#### 2.4.2. Capacity Measures

##### *Standardized Clinical Assessments—Body Function and Activity Domains of* *ICF-CY*

Standardized clinical assessments were used to measure changes in UE capacity pre- and post-HABIT in a controlled laboratory setting [15]. The AHA assesses the affected hand function and bimanual coordination in children with UCP [31]. An improvement of 5 units is considered clinically meaningful [32]. JHFT (reliability; interrater = 0.94, test–retest = 0.91 [33]) and NHPT (reliability; interrater = 0.99, test–retest = 0.81 [34]) measure unimanual dexterity and speed. BBT (reliability; interrater = 0.99, test–retest = 0.85 [35]) assesses the unimanual speed.

### 2.5. Statistical Analysis

Data were analyzed using IBM Statistical Package for Social Sciences Version 28.0.0. Data are presented as mean ± SD for continuous variables and *n* (%) for categorical variables. The intensity of UE movement characteristics during 5 days of HABIT were quantified using descriptive statistics for UR, MR, BM, MA, AV, and AAC. Repeated measures analysis of variance (ANOVA) was used with time (five days) as a within-group variable to determine variability in training using UR, MR, BM, MA, AV, and AAC. Considering the repetitive measurements, the significance level for the ANOVA was set at *p* value ≤ 0.01 using the Bonferroni method. Capacity and performance outcomes were assessed for normality using the Shapiro–Wilk test. Pre- and post-HABIT changes in the capacity and performance measures were assessed using a paired *t*-test for all the variables, except for MR and BBT scores. The Wilcoxon signed rank test was used to analyze changes in MR and BBT scores pre- and post-HABIT as the data violated the assumption of normality. The significance level was set at a *p* value = 0.05 for the paired *t*-test.

## 3. Results

Twenty-eight participants were enrolled, and twenty-six completed the study intervention. However, data were analyzed for only 25 participants due to incomplete accelerometer data from one participant. There were no adverse events reported during HABIT. Power was derived based on Goodwin et al.’s [21] study and computed using G*Power [36]. To detect the mean difference of 0.25 (μ1 = 1.36, μ2 = 1.61; SD1 = 0.12, SD2 = 0.21) in the primary outcome use ratio (UR), a total of 26 participants provides 94% power to detect an effect size of 1.46 at a significance level of 0.05. The sample size was calculated based on a two-sided *t*-test.

### 3.1. Characteristics of Bimanual Movement Intensity during HABIT

Accelerometer metrics, UR, MR, BM, MA, AV, and AAC, representing the bimanual movement characteristics during HABIT days, are summarized as descriptive statistics in Table 2. All the children participated in 30 h of HABIT training across five days. Overall, during HABIT, the affected UE use was 47.26% as compared to 53.73% of the less affected UE use.

(1)Magnitude ratio (MR): The average MR across five days of HABIT was −0.56 ± 0.26 (range: −0.97 to 0.05, Figure 3b). There was no significant main effect of time (F (4,96) = 1.688, *p* = 0.159) for MR.(2)Bilateral Magnitude (BM): The average BM across five days of HABIT was 167.25 ± 39.83 (range: 101.98–267.51, Figure 3c). There was no significant main effect of time (F (4,96) = 1.923, *p* = 0.113) for BM.(3)Median acceleration (MA): The average MA across five days of HABIT was 56.99 ± 21.21 (range: 28.49–115.07, Figure 3d). There was no significant main effect of time (F (4,96) = 2.004, *p* = 0.1) for MA.(4)Acceleration variability (AV): The average AV across five days of HABIT was 110.27 ± 18.63 (range: 69.07–143.05, Figure 3e). There was a significant main effect of time (F (4,96) = 3.666, *p* = 0.008). Table 3 shows significant post hoc analysis results using Bonferroni multiple comparisons for acceleration variability across five days of HABIT.(5)Affected extremity activity counts (AAC): The average daily number of affected UE accelerations during 6 h across five days of HABIT were (mean ± SD) 15,399 ± 2477 (range: 9863–20,057 counts) (Figure 3f). The total affected UE accelerations reflecting UE use (sum of the means of daily affected UE accelerations) during 30 h of HABIT was 76,997 movements. There was a significant main effect of time (F (4,96) = 2.633, *p* = 0.03) for the AAC. Table 3 shows significant post hoc analysis results using Bonferroni multiple comparisons for AAC across five days of HABIT.

### 3.2. Pre- and Post-HABIT Change in Upper Extremity Performance Measures: Activity and Participation

There was a significant improvement in UR (*p* = 0.002, Figure 4a), MR (*p* = 0.018, Figure 4b), bilateral magnitude (*p* = 0.006, Figure 4c), median acceleration (*p* = 0.002, Figure 4d), and acceleration variability (*p* = 0.024, Figure 4e) post-HABIT. These findings indicate that 30 h of HABIT enhanced children’s use of the affected arm in terms of movement symmetry, magnitude, and variability. Appendix A shows exemplary data from a study participant with pre- and post-HABIT changes in these performance measures. Appendix A shows the inter- and intra-individual differences in accelerometer-derived variables for all the participants across time points.

### 3.3. Pre- and Post-HABIT Change in Capacity Measures of the Affected Upper Extremity: Body Function and Activity

There was significant improvement in the mean scores of the AHA (*p* = 0.001, Figure 5a), the JHFT (*p* = 0.001, Figure 5b), the BBT (*p* = 0.002, Figure 5c), and the NHPT (*p* = 0.011, Figure 5d) from pre- to post-HABIT. The mean scores of AHA and BBT exceeded the minimal clinically important difference (MCID) of five logit scores [32] and two blocks [37]. The mean score of the JHFT was very close (53.4 s) to the MCID of 55 s [37]. These findings indicate that post-HABIT, children showed an increase in bimanual coordination, dexterity, and speed of the affected hand use.

## 4. Discussion

Our primary aim was to quantify the characteristics of bimanual movement intensity during 30 h of HABIT utilizing bilateral wrist-worn accelerometers. Our findings illustrate that the standard accelerometer-derived variables can quantify the contribution of the affected UE to bimanual activities and hence can provide an objective metric for the intensity of bimanual movements practiced during HABIT. Moreover, our results indicate that children performed a total of 76,997 accelerations with their affected UE during HABIT. Our secondary aim was to examine gains in bimanual performance (activity and participation in a real-world environment) and capacity (body function and activity in a clinical setting) post-HABIT using accelerometer-derived metrics and standardized outcomes, respectively. We found significant gains in real-world performance of bimanual activities post-HABIT, suggesting improved activities and participation in the child’s natural, real-world environment. Furthermore, consistent with prior studies, our results indicate improvements in UE capacity following HABIT. Overall, this is the first study that utilized wearable technology to quantify the intensity of HABIT and demonstrated that accelerometers can objectively quantify bimanual movement characteristics reflecting the intensity of UE use during HABIT. Moreover, 30 h of HABIT has the potential to improve the UE capacity as well as real-world bimanual performance in children with UCP.

Time is a dominant measure used to define the intensity of arm use during intensive therapies [4,5]. However, time does not indicate the actual number of movements or movement characteristics performed during a particular session [9,38]. In this study, we overcame this limitation using accelerometers and demonstrated a more accurate method to objectively quantify UE movement characteristics during HABIT. UR and MR signify the contributions of the affected relative to the less affected UE considering the duration and magnitude (range of movement) during bimanual activities. Children in our study attained an average UR of 0.90 during HABIT, which was 22% higher than pre-training. The UR being close to 1 suggests that there was nearly equal use of both UEs during HABIT. The average MR during HABIT was −0.56, which was 64.6% higher than pre-training. The value of MR moved substantially closer to 0, indicating a greater magnitude of the affected UE during training. Likewise, the average BM reflecting the combined magnitudes of accelerations from both UEs during HABIT was 167.25, which was 58.8% higher than pre-training. The average MA and AV reflecting mean accelerations of the affected UE and variability of accelerations also increased (MA = 56.9, AV = 110.3) noticeably during HABIT by 154.36% and 54.4% compared to pre-training. Collectively, these results indicate greater symmetry in UE use and higher affected arm use during HABIT. These improvements can be attributed to the intensive nature of bimanual, task-specific activities incorporated in HABIT. Notably, the UR and MR attained during training were comparable to those reported in accelerometer studies in typically developing children of 0.96 and −0.28, respectively, which suggests that the bimanual activities incorporated in HABIT were comparable with the amount of typical bimanual activities [21,26]. Overall, our findings provide preliminary evidence for using accelerometers to quantify movement characteristics during HABIT. These findings could serve as the foundation for future studies to understand the relationship between accelerometer metrics and motor outcomes in children with UCP.

Affected UE activity counts (AAC) during 6 h of HABIT ranged from 14,714 to 16,160, and a total of nearly 76,997 accelerations occurred during 30 h of HABIT. The use of repetitions to quantify intensity is limited in rehabilitation research and clinical practice. Some studies on stroke survivors have investigated repetitions of UE training by observing video recordings of therapy sessions [14,39]. Lang and colleagues used repetitions to quantify the intensity of upper limb exercise during stroke rehabilitation and inpatient hospital stays [40]. These observational studies, however, focused on routine therapy sessions, which are invariably of shorter duration. Although this is the gold standard method for monitoring repetitions, manual coding by human observers could be exceedingly strenuous, especially for extended hours of therapy such as HABIT. Despite the differences regarding the inability of accelerometers to isolate purposeful movements, UE movement characteristics derived from accelerometer metrics could still provide clinically relevant data about UE use during prolonged hours of training. Moreover, previous studies have found an agreement between the affected UE activity counts and human-observed purposeful repetitions during group [41] and individual therapy [39] sessions, suggesting that accelerometer measures have concurrent validity. As a result, this study is unique in two ways: (1) for the first time, we provide objective data on bimanual movement characteristics reflecting the intensity of arm use during HABIT; and (2) we were able to quantify the number of affected UE activity counts during 30 h of HABIT, which, while an overestimation, could still be considered a key component to influence motor learning.

Our results demonstrated significant improvements on all standardized clinical tests, reflecting enhanced UE capacity and suggesting improvements in body function and activity measured in a constrained clinical environment post-HABIT. Notably, the AHA scores exceeded the minimal clinically important (MCID) difference of five units [32]. Children were able to transfer a greater number of blocks (>2 blocks compared to pre-training, MCID = 2 [37]) with their affected UE (12% increase in BBT) and complete the NHPT (21.04% faster) and JHFT (19.35% faster, pre-to-post-HABIT difference = 53.4 s, very close to MCID of 55 s) [37] in less time, indicating greater speed and dexterity post-HABIT. These findings are consistent with previous studies that demonstrated an increase in hand capacity following intensive therapies [18].

Our study results also indicate gains in UE performance following HABIT, which indicates enhanced activities and participation in a natural, real-world environment in these participants [15]. Children demonstrated increases in UR, MR, and BM of 6.85%, 34.81%, and 15.84%, respectively, indicating greater symmetry and contribution of the affected UE in terms of hours of use and magnitude of real-world bimanual activities. Additionally, post-HABIT, MA and AV increased by 39.56% and 9.48%, respectively, which indicates an increase in the affected UE speed and variations in movement speed following HABIT. However, our accelerometer measures also revealed a significant degree of variability at pre- and post-HABIT time points, aligning with previous studies that employed similar methods [21,27]. This variability can be attributed to the heterogeneity of our study population, including differences in age, gender, and functional level as measured by MACS. Further analysis of the individual profile plots depicting inter-individual differences (Appendix A) revealed that a few participants with MACS level III exhibited higher UR (>0.8) at baseline. This could be due to involuntary and mirror movements in their more affected UE, resulting in higher accelerometer readings indicating increased affected UE use. Additionally, the profile plot of MR (Appendix A) indicated that the changes in the magnitude of movement of the more affected UE following HABIT were significantly driven by three participants. These three participants were relatively older (≥15 years) compared to younger children and likely had higher motivation for high-amplitude activities, such as overhead catch and throw with a soccer ball or hitting a baseball, etc., which plausibly led to greater improvements in MR post-HABIT. The profile plots for BM, MA, and AV (Appendix A) revealed similar trends: participants with MACS level III, who initially had lower baseline scores, exhibited more pronounced improvements after undergoing HABIT compared to those with MACS levels I and II. Similar to MR, the profile plots for BM and MA indicated that a small subset of participants with MACS level III, who were comparatively older demonstrated pronounced changes post-HABIT. Despite the observed variability, which is typical in clinical populations, collectively, our study findings demonstrated improvements in real-world bimanual performance, which contradict the findings of previous studies that utilized accelerometers to assess UE performance gains post-CIMT [21,27]. The limited gains reported in those studies could be due to the lack of a bimanual training component in the CIMT approach. We believe the intensity of HABIT administered in our study, as seen by improved UE movement characteristics, was potentially adequate to drive changes in UE capacity beyond a specific threshold required to produce a change in UE performance.

Study Limitations: We recognize a few study limitations and propose future study directions. First, wrist-worn accelerometers capture only arm and forearm accelerations (gross motor function), but they are limited in capturing finger accelerations (fine motor function). Thus, future studies could use finger-worn inertial sensors to quantify fine movements in this population. Second, although we attempted to perform bimanual activities during HABIT training to achieve purposeful movements, a part of our data may contain non-purposeful movements occurring during non-therapy time, such as normal walking, washroom breaks, etc. Therefore, caution should be used when interpreting the results of this study. Third, we did not measure long-term retention of the UE performance gains in this study. We suggest that future studies address this limitation by conducting follow-up assessments to determine the persistence of immediate gains in performance, which could provide insight into the retention and transfer components of motor skill learning. Finally, the generalizability of our study findings could be limited due to heterogeneous study participants and a lack of a control group.

## 5. Conclusions

Accelerometers can be used to quantify the movement characteristics of UE during HABIT, which could provide an objective measure regarding the intensity of UE use. Thirty hours of HABIT has the potential to improve UE function in real-world bimanual activities, indicating improvements in activities and participation in the natural environment, and to enhance the speed and dexterity of the affected UE, indicating improvements in body function and activity in a clinical environment. Overall, the accelerometer is a valuable tool for clinicians to conveniently quantify the different aspects of UE movements and monitor in-clinic as well as real-world improvements in UE use in children with UCP.

## Figures and Tables

**Figure 1 behavsci-13-00681-f001:**
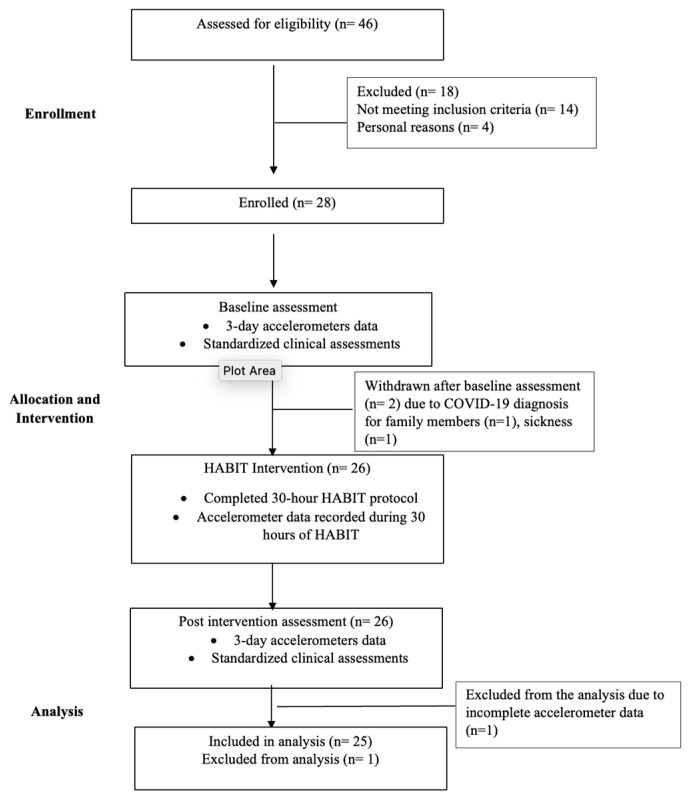
CONSORT flow chart.

**Figure 2 behavsci-13-00681-f002:**
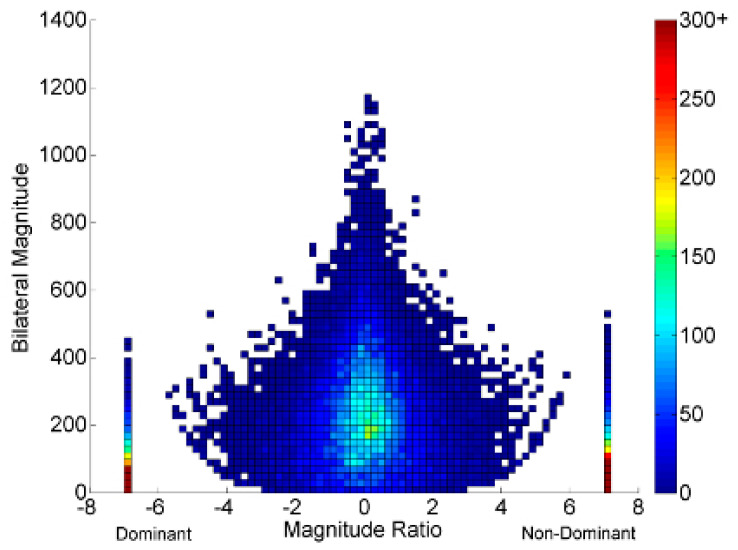
A representative example of a density plot, which is a graphical representation of accelerometer data obtained from both upper extremities (UE) of a typically developing child. The plot shows accelerations on a second-by-second basis recorded over a total wear time of three days, both pre- and post-HABIT in our study. The *x*-axis represents the magnitude ratio, indicating the contribution of each UE to the task, while the *y*-axis represents the overall intensity or magnitude of movement. The right and left halves of the plot represent the use of the right and left UEs, respectively. The dots visible in the graph indicate the counts or number of movements (accelerations) performed by each UE. The large color bar scale on the right of the plot displays the frequency of movements, with brighter colors indicating greater frequencies and vice versa. The plot is noticeably symmetrical, indicating that the typically developing child used both UEs equally in terms of hours, which is called the use ratio. The magnitude ratio indicates the contribution of the affected UE relative to the less affected UE in terms of the intensity of movements. The dots seen in both halves of the plot appear at similar heights, suggesting a symmetrical magnitude of movements with both UEs in a typically developing child. Bilateral magnitude is the overall intensity of both UEs, which is determined by the height of the density plot. A taller density plot indicates better bilateral magnitude. Median acceleration and acceleration variability are measures reflecting the mean accelerations and variance in the accelerations of the affected UE, which are shown in only one half of the density plot, depending on the affected side.

**Figure 3 behavsci-13-00681-f003:**
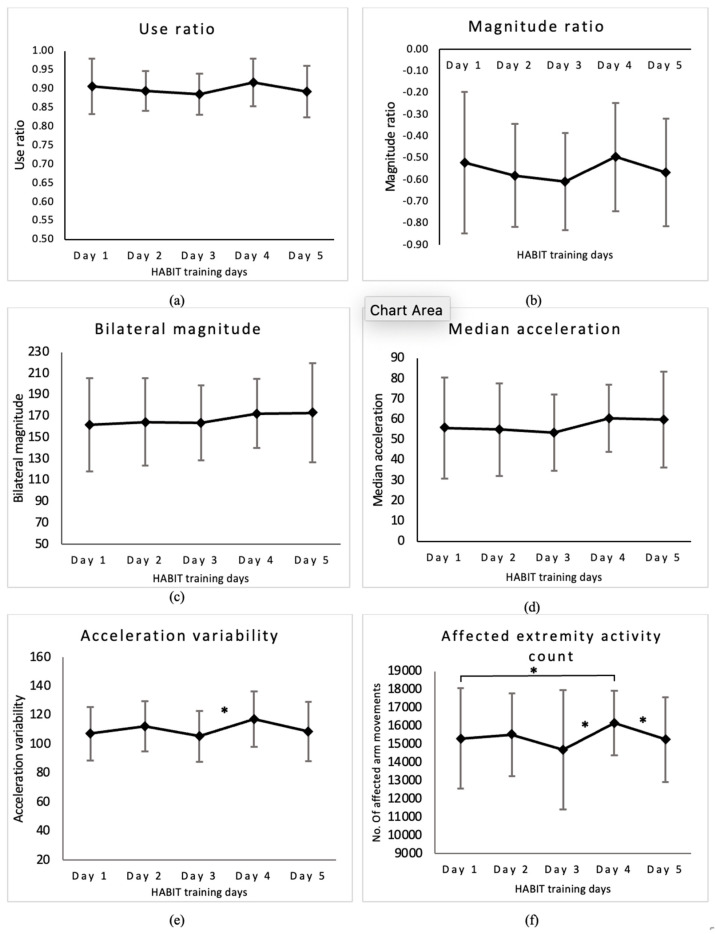
HABIT intensity across 5 days of training using accelerometer-derived variables. Values are means ± SD for each day of HABIT: (**a**) use ratio, (**b**) magnitude ratio, (**c**) bilateral magnitude, (**d**) median acceleration, (**e**) acceleration variability, and (**f**) affected extremity use count. Variability is observed in all the accelerometer variables during the five training days. * denotes a significant *p* value at α = 0.05.

**Figure 4 behavsci-13-00681-f004:**
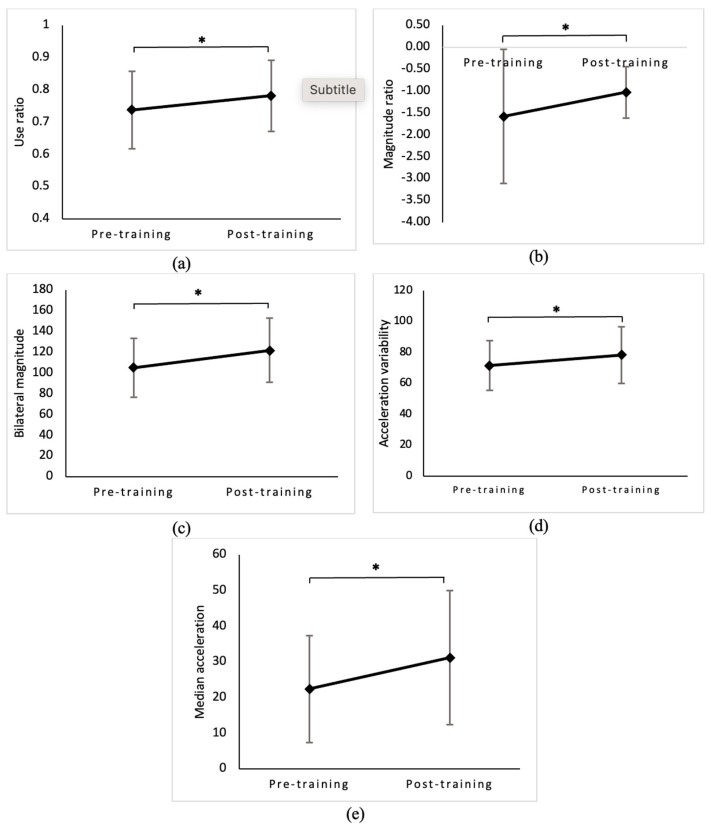
Comparison of differences in the mean scores of accelerometer-derived variables to assess performance gains (activity and participation) pre- and post-HABIT training. Values reported are means ± SD or median (range) as determined by the distribution of data during each assessment time point. Pre-training refers to baseline assessment, and post-training refers to assessment within one week following HABIT. There was a significant change in the average (**a**) use ratio, (**b**) magnitude ratio, (**c**) bilateral magnitude, (**d**) acceleration variability, and (**e**) median acceleration from pre- to post-HABIT. * indicates a significant p value at α = 0.05.

**Figure 5 behavsci-13-00681-f005:**
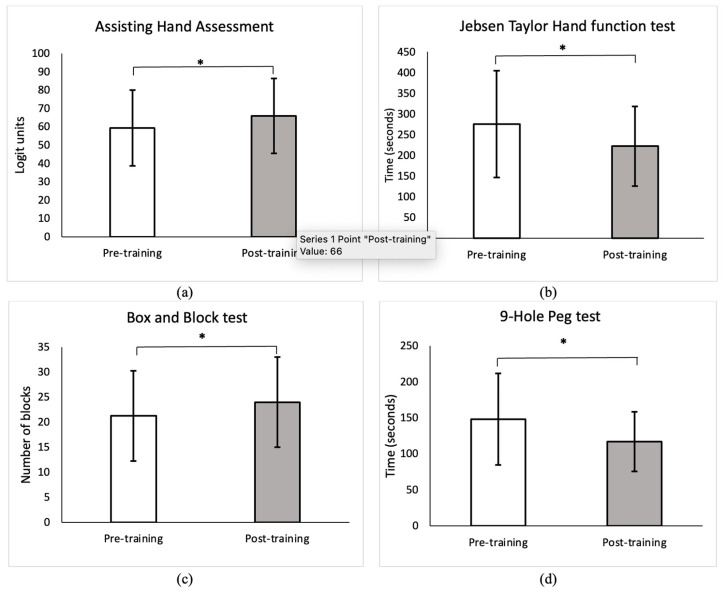
Comparison of differences in the mean scores of capacity (body function and activity) measures pre- and post-HABIT. Values reported are means ± SD as determined by the distribution of data during each assessment time point. Pre-training refers to baseline assessment, and post-training refers to assessment within one week following HABIT. There were significant changes seen in the mean scores of (**a**) Assisting Hand Assessment (>MCID of 5 logit scores), (**b**) Jebsen Taylor Hand Function Test [very close (53.4 s) to MCID of 55 s], (**c**) Box and Block Test (>MCID of 2 blocks), and (**d**) Nine Hole Peg Test pre- vs. post-training. Children demonstrated significant improvement in bimanual coordination, speed, and dexterity as reflected by a greater AHA score, faster speeds in JHFT and NHPT, and a greater number of blocks transferred in BBT, respectively, during post-training assessment compared to baseline. * indicates a significant *p* value at α = 0.05.

**Table 1 behavsci-13-00681-t001:** Demographic details of the participants.

Characteristics	Participants
	Children with Unilateral Cerebral Palsy (*n* = 25)
Sex, *n* (%)	
Male	18 (72)
Females	7 (28)
Age, mean (SD)	11.20 (3.59)
Side of hemiplegia, *n* (%)	
Left	10 (40)
Right	15 (60)
Race, *n* (%)	
White	21 (84)
African American	1 (4)
Asian	3 (12)
Multiracial	-
MACS Level, *n* (%)	
I	1 (4)
II	9 (36)
III	15 (60)

MACS indicates Manual ability classification system.

**Table 2 behavsci-13-00681-t002:** Descriptive statistics of accelerometer-derived variables across five days of HABIT.

Accelerometer Variables	Minimum	Maximum	Mean	Std Deviation
Use Ratio				
Day 1	0.81	1.13	0.91	0.07
Day 2	0.80	0.99	0.89	0.05
Day 3	0.78	1.00	0.89	0.06
Day 4	0.81	1.08	0.92	0.06
Day 5	0.77	1.05	0.89	0.07
Average	0.79	1.05	0.90	0.06
Magnitude Ratio				
Day 1	−0.96	0.69	−0.52	0.33
Day 2	−1.09	−0.13	−0.58	0.24
Day 3	−1.03	−0.27	−0.61	0.22
Day 4	−0.91	−0.01	−0.50	0.25
Day 5	−0.87	−0.04	−0.57	0.25
Average	−0.97	0.05	−0.56	0.26
Bilateral Magnitude				
Day 1	93.78	268.94	161.99	43.80
Day 2	102.43	296.16	164.69	41.04
Day 3	98.53	240.82	163.87	35.36
Day 4	114.13	257.18	172.56	32.44
Day 5	101.02	274.46	173.16	46.50
Average	101.98	267.51	167.25	39.83
Median Acceleration				
Day 1	25.00	117.14	55.85	24.82
Day 2	28.44	134.99	55.04	22.69
Day 3	25.30	101.24	53.56	18.64
Day 4	37.16	104.35	60.61	16.50
Day 5	26.57	117.63	59.89	23.40
Average	28.49	115.07	56.99	21.21
Acceleration Variability				
Day 1	73.44	142.26	107.28	18.65
Day 2	79.14	140.00	112.29	17.26
Day 3	63.44	132.51	105.49	17.63
Day 4	73.08	158.11	117.40	19.09
Day 5	56.25	142.36	108.89	20.53
Average	69.07	143.05	110.27	18.63
Affected Use count				
Day 1	10,198	20,283	15,325	2745
Day 2	11,100	20,823	15,531	2280
Day 3	3790	19,760	14,714	3275
Day 4	12,903	19,996	16,160	1767
Day 5	11,322	19,425	15,267	2317
Average	9863	20,057	15,399	2477

Use ratio (UR): The average use ratio across five days of HABIT was 0.90 ± 0.06 (range: 0.79–1.00, Figure 3a). There was no significant main effect of time (F (4,96) = 1.873, *p* = 0.121) for UR.

**Table 3 behavsci-13-00681-t003:** Pairwise comparison of the accelerometer metrics between different HABIT training days.

Bonferroni Pairwise Comparison								
**Variables**	**Mean**	**vs.**	**Mean**	**Significance**	**Mean Difference**	**Standard Error of Difference**	**Lower Bound**	**Upper Bound**
**Acceleration variability**	Day3	vs.	Day4	0.001 *	−11.9	3.6	−19.3	−4.6
**Affected extremity activity counts**	Day1	vs.	Day5	0.04 *	−835.3	385.0	−1629.9	−40.7
	Day3	vs.	Day5	0.024 *	−1445.6	601.9	−2687.9	−203.2
	Day4	vs.	Day5	0.036 *	893.5	402.7	62.5	1724.6

Post hoc analysis results with Bonferroni multiple comparisons between training days; * indicates a significant *p* value at α = 0.05.

## Data Availability

The data presented in this study are available on request from the corresponding author.

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
