# Peer review of "Bimanual Movement Characteristics and Real-World Performance Following Hand–Arm Bimanual Intensive Therapy in Children with Unilateral Cerebral Palsy"

_behavsci, 2023, doi:10.3390/bs13080681_

Round 1
Reviewer 1 Report (Previous Reviewer 1)
Dear authors
the revisions are very adequate and the information is updated and improved. good the authors included the statistician.
the paper is fully acceptable for publication for me.
Well done
Author Response
Please see the attachment.

Reviewer 2 Report (New Reviewer)
The article “Bimanual movement characteristics and real-world performance following Hand Arm Bimanual Intensive Therapy in children with unilateral cerebral palsy” contributes to understanding the effects of HABIT on children with UCP while utilizing innovative approaches, such as wrist-worn accelerometers. It shows the possibility of using accelerometers and provide a comprehensive understanding of movement characteristics and the effectiveness of HABIT. While the study demonstrates significant contributions, there could be potential improvements that can be considered:
Point 1. The accelerometers used in the study had a low sampling rate of 30 Hz, which might hinder the capturing of fine motor functions, crucial for a complete understanding of movement characteristics.
Point 2. The generalizability could be limited due to the lack of a control group.
Point 3. The demographic details provided include parental education, a factor that does not seem to have a direct connection to the research. It might be beneficial to clarify the relevance of this information, or to exclude it if it's found to be unrelated to the study’s objectives.
Overall, the article is well-structured, with the introduction, methodology, and results sections being particularly well-designed.
Future studies could include control group to provide a more comprehensive comparison and would be interesting to see sustainability of the HABIT benefits in a long-term follow-up utilizing more advanced accelerometer systems or other devices to track finger movements and provide a more detailed assessment of fine motor function.
In conclusion, this article represents a valuable contribution to the field, offering innovative methods and insights while also pointing towards exciting avenues for future research.
Author Response
Please see the attachment.

This manuscript is a resubmission of an earlier submission. The following is a list of the peer review reports and author responses from that submission.
Round 1
Reviewer 1 Report
Bimanual movement characteristics and real-world performance following Hand Arm Bimanual Intensive Therapy in children with cerebral palsy
I thank the authors for this clinically relevant and useful paper, underlining the importance of HABIT.
The paper is well written.
I have mainly some question to improve the paper.
You explained well the capacity and performance, it would be good to embed the purpose of the paper in the ICF-CY and explaining how the activity monitoring is helping, indicating, activity levels and participation.
Use the ICF-CY to label all outcome measures and their relation with the accelerometry.
Methodology
a. Measuring activity by wrist amount of movements or overall arm movement during the HABIT on 5 days is well done. Please present all individual curves of the outcomes to profile the candidates on all outcome and to discus about the intra individual differences and inter individual differences on each test moment.
b. You present only means and sd, showing a huge variation, however not explaining this variation. The individual plots with this high measurement repetition is relevant to show profiles of changes. It will enrich the discussion
c. It is expected that in CIMT and HABIT both arms will increase the amount of movement, due to the intensity and goal of the program. I am confused that no longer term measurements are performed after 1-2-3 month, to see if the amount of movement has really changed
d. A very essential comment is that characteristics of activity movements are presented as total. No specific attention has been presented on goal directed movements – related to tasks- and no goal directed movements. What does the increase tell us if we do not know if they are due to the task specific approach or just following many activities using the upper limb.
e. The relation between real world improvement in the program is doubtful, because it is still an in clinic approach, with a manipulated- adapted environment, but not the school, house, sports, play environment. Please embed this topic into your discussion, because you did not measure during a regular day at school, at home etc…
I advise the help of a statistician to look in the analysis, based on the repetitive measurements, the choice of ANOVA and the impact on p value, sample size.
Best
Author Response
We sincerely thank the reviewers for their valuable feedback, which helped in further strengthening the rigor and impact of our work. Below is point-by-point response to the reviewers’ comments.
Reviewer 1.
- You explained well the capacity and performance, it would be good to embed the purpose of the paper in the ICF-CY and explaining how the activity monitoring is helping, indicating, activity levels and participation. Use the ICF-CY to label all outcome measures and their relationship with the accelerometry.
Response: Thank you for your feedback. We have defined performance and capacity domains as per WHO’s ICF-CY guidelines in the introduction and included the references in the methods and discussion sections of the revised manuscript (please see lines 56-57, 72-79, 83, 182, 187, 234, 309-310, 327-328, 357-359, 361-362, 416-418, 426-428).
- Measuring activity by wrist amount of movements or overall arm movements during the HABIT on 5 days is well done. Please present all individual curves of the outcomes to profile the candidates on all outcome and to discuss about the intra individual differences and inter individual differences on each test moment.
Response: Thank you for your suggestion. We have included the individual data plots for all accelerometer variables representing inter- and intra-individual differences in study participants (please see supplementary material 3). We also strengthened our discussion by thoroughly explaining these inter- and intra-individual differences (please see lines 433-456).
- You present only means and SD, showing a huge variation, however not explaining this variation. The individual plots with this high measurement repetition are relevant to show profiles of changes. It will enrich the discussion.
Response: Thank you for your comment. We have included the individual data plots (please see supplementary material 3) and incorporated the potential reasons for the high variability in accelerometer outcomes in the discussion section (please see lines 433-456).
- It is expected that in CIMT and HABIT both arms will increase the amount of movement, due to the intensity and goal of the program. I am confused that no longer term measurements are performed after 1-2-3 month, to see if the amount of movement has really changed.
Response: We appreciate your comment. The primary aim of our study was to assess the upper extremity (UE) movement characteristics during Hand-Arm Bimanual Intensive Therapy (HABIT) and to evaluate immediate gains in UE performance using an objective method such as accelerometry. While long-term follow-up to assess the retention effects would be valuable, we first need to demonstrate whether HABIT enhances bimanual performance. Since our results have now demonstrated that HABIT has potential to enhance bimanual performance in the real-world environment, our next steps are to assess the retention effects. We acknowledged the importance of follow-up assessments in the discussion section (please see lines 469-473).
- A very essential comment is that characteristics of activity movements are presented as total. No specific attention has been presented on goal directed movements – related to tasks- and no goal directed movements. What does the increase tell us if we do not know if they are due to the task specific approach or just following many activities using the upper limb.
Response: HABIT was administered in accordance with established protocols, which involved implementing structured, task-specific, age-appropriate bimanual gross and fine motor tasks within a playful context (please see supplementary material 1). The activities were tailored to each participant based on pre-training assessments, as well as the bimanual goals identified by both parents and children. According to our protocol, we are confident that the majority of the activity counts recorded during therapy were purposeful and directed towards the set goals. However, we do acknowledge that there are some limitations in using accelerometers as a tool to quantify UE activity, as they measure a combination of purposeful and non-purposeful movements performed during training. Furthermore, studies indicate a high to moderate correlation between the affected UE activity counts and human-observed purposeful repetitions (please see lines 408-410). Therefore, although, few activities during HABIT may involve non-goal-directed movements such as hand movements during walking in the hallway, during break time etc., majority of the movements practiced involves purposeful movements as a part of the HABIT protocol. We have addressed the involvement of some non-purposeful movements as one of the limitations of our study (please see lines 465-469).
- The relation between real world improvement in the program is doubtful because it is still an in-clinic approach, with a manipulated- adapted environment, but not the school, house, sports, play environment. Please embed this topic into your discussion, because you did not measure during a regular day at school, at home etc.…
Response: As explained in our methods section, we assessed upper extremity (UE) performance gains through accelerometer measurements over three full consecutive days before and after HABIT, which captured their activities in home, school, and play environment. This approach was designed to provide an accurate representation of the effects of HABIT in a natural environment. The data in Figure 4 represents the accelerometer data collected over three consecutive days (72 hours).
We have incorporated the changes in the methods section to clarify our accelerometry data acquisition procedure (please see lines 170-174).
- I advise the help of a statistician to look in the analysis, based on the repetitive measurements, the choice of ANOVA and the impact on p value, sample size.
Response: Thank you for your input. We consulted a biostatistician to review the analysis, who confirmed that using a repeated measures ANOVA to compare the differences in accelerometer-derived variables over the course of 5 days of HABIT was appropriate. The biostatistician is now a co-author in the manuscript, who also approved that the illustration (Figure 3) depicting repeated measures ANOVA is appropriate. Furthermore, the biostatistician suggested to adjust the significance level of the p-value to 0.01 considering multiple comparisons in the repeated measures design and hence, to reduce the probability of obtaining a false positive result (type I error). Therefore, we adjusted the p value for the repeated measures ANOVA using Bonferroni method by dividing 0.05 with 5 (since we have 5 comparisons i.e., 5 HABIT training days), thereby setting the new significance level as ≤ 0.01. This adjustment makes the criterion for statistical significance more stringent to counteract the increased probability of false positives. We re-analyzed the data per new statistical significance and incorporated the necessary changes in the statistical analysis section of the main text file (please see lines 249-250) and revised Table 3 and Figure 3). We also included the power analysis (please see lines 260-263).
Reviewer 2 Report
The main problem I see with this study is the sample size, only 25 participants, so with this limitation it is difficult to extrapolate the results to the general population.
Furthermore, in methodology the authors state that this study is a secondary analysis of a double-blind, randomized controlled trial (NCT05355883), what do the authors mean by this? The study has already been published and these data are a part of that study? Data from a study can only be published once.
On the other hand, the bibliography needs to be updated, as there are no updated references. Furthermore, the references do not follow the indications of the journal's regulations.
Author Response
Reviewer 2.
- The main problem I see with this study is the sample size, only 25 participants, so with this limitation it is difficult to extrapolate the results to the general population.
Response: Thank you for your feedback. We have included the sample size calculation in the main text file (please refer to lines 260-263). Twenty-five participants provide 94% power to detect pre- and post-HABIT changes in the primary outcome measure, use ratio.
Furthermore, Goodwin et al. and Coker-Bolt et al. conducted previous studies on real-world performance after Constrained Induced Movement Therapy using accelerometers, involving seven and twelve children with UCP, respectively. In comparison, our study sample is larger, and we are sufficiently powered to detect an effect size of 1.46 at a 0.05 significance level.
References:
- Goodwin, B. M.; Sabelhaus, E. K.; Pan, Y. C.; Bjornson, K. F.; Pham, K. L. D.; Walker, W. O.; Steele, K. M. Accelerometer Measurements Indicate That Arm Movements of Children with Cerebral Palsy Do Not Increase after Constraint-Induced Movement Therapy (CIMT). Am J Occup Ther. 2020, 74 (5): :7405205100p1-7405205100p9.
- Coker-Bolt, P.; Downey, R. J.; Connolly, J.; Hoover, R.; Shelton, D.; Seo, N. J. Exploring the Feasibility and Use of Accelerometers before, during, and after a Camp-Based CIMT Program for Children with Cerebral Palsy. J Pediatr Rehabil Med. 2017, 10 (1), 27–36.
- Furthermore, in methodology the authors state that this study is a secondary analysis of a double-blind, randomized controlled trial (NCT05355883), what do the authors mean by this? The study has already been published and these data are a part of that study. Data from a study can only be published once.
Response: Thank you for your comment. It was our oversight to report the study as a secondary analysis. The current study is an ancillary analysis of the ongoing clinical trial (NCT05355883). The complete data set from the parent study has not been analyzed or published elsewhere yet. We collected additional data during parent study using accelerometers, which did not interfere with the primary objectives of the parent clinical trial. The results of the ancillary analyses are worth sharing since these are novel findings.
We have replaced the word ‘secondary analysis’ with ‘ancillary analyses’ with appropriate citation in the methods section (please see lines 119-120).
Reference:
For ancillary studies, consider NIH definitions carefully. (2021, October 6). National Institute of Allergy and Infectious Diseases (NIAID). https://www.niaid.nih.gov/grants-contracts/ancillary-studies-definitions
- On the other hand, the bibliography needs to be updated, as there are no updated references. Furthermore, the references do not follow the indications of the journal's regulations.
Response: We included the updated references and formatted the bibliography as per journal’s guidelines in the revised manuscript draft (please see lines 521-623).